# The Inhibitory Effects of Hydroxytyrosol, α-Tocopherol and Ascorbyl Palmitate on Lipid Peroxidation in Deep-Fat Fried Seafood

**DOI:** 10.3390/antiox12040929

**Published:** 2023-04-14

**Authors:** Audrey Yue Vern Theah, Taiwo O. Akanbi

**Affiliations:** School of Environmental and Life Sciences, College of Engineering, Science and Environment, The University of Newcastle (UON), Brush Road, Ourimbah, NSW 2258, Australia

**Keywords:** squid, hoki, prawn, sunflower oil, hydroxytyrosol, ascorbyl palmitate, tocopherol, oxidation, deep-fat frying

## Abstract

This study aimed to investigate the inhibitory effects of hydroxytyrosol, α-tocopherol and ascorbyl palmitate on lipid peroxidation in squid, hoki and prawn during deep-fat frying and refrigerated storage. Fatty acid analysis using gas chromatography (GC) showed that the seafood had a high omega-3 polyunsaturated fatty acid (n-3 PUFAs) content, including docosahexaenoic acid (DHA) and eicosapentaenoic acid (EPA). The total content of n-3 fatty acids in their lipids was 46% (squid), 36% (hoki) and 33% (prawn), although they all had low lipid contents. The oxidation stability test results showed that deep-fat frying significantly increased the peroxide value (POV), *p*-anisidine value (*p*-AV) and the value of thiobarbituric acid reactive substances (TBARS) in squid, hoki and prawn lipids. Meanwhile, antioxidants delayed the lipid oxidation in fried seafood and sunflower oil (SFO) used for frying, albeit in different ways. The least effective of all the antioxidants was α-tocopherol, as the POV, *p*-AV and TBARS values obtained with this antioxidant were significantly higher. Ascorbyl palmitate was better than α-tocopherol but was not as effective as hydroxytyrosol in suppressing lipid oxidation in the frying medium (SFO) and in the seafood. However, unlike the ascorbyl palmitate-treated oil, hydroxytyrosol-treated oil could not be used for multiple deep-fat frying of seafood. Hydroxytyrosol appeared to be absorbed in the seafood during multiple frying, thus leaving a low concentration in the SFO and making it susceptible to oxidation.

## 1. Introduction

Deep-fat frying is the submerging of foods in a high-temperature fat medium until cooked or desired. It involves frying food in hot vegetable oil at a temperature between 130 and 200 °C for about 2–10 min (depending on the food) [1]. High temperature increases the activities between food components such as proteins andcarbohydrates, the dehydration of the crust and the oil uptake [2]. This process may initiate the thermo-oxidation and polymerization of fatty acids in the food and the frying oil, which is the heating medium. Additionally, since the oil gets absorbed by the food during deep frying, the quality of dietary fat in the food may be affected because of the heat transfer from the frying medium to the food and vice versa.

Generally, seafood such as fish and shellfish contain lipids that are high in omega-3 (n-3) polyunsaturated fatty acids (PUFAs), such as docosahexaenoic acid (DHA) and eicosapentaenoic acid (EPA), and their health benefits have been widely reported. For instance, dietary EPA and DHA have been found to control and regulate several cellular activities in the human body, leading to the development of organs and the prevention of cardiovascular diseases such as arrhythmias, atherothrombosis and inflammations [3]. Also, the regular consumption of omega-3-rich seafood has been linked with a reduced risk of dementia and Alzheimer’s disease [4]. However, the intended biological benefits of these fatty acids may be compromised and become potentially harmful to the body when consumed in an oxidized form [5].

Lipid oxidation is a chain reaction involving free radicals, peroxides, oxidized volatile products and small molecular substances. The susceptibility of lipids to oxidation is linked with the degree of the unsaturation of the fatty acids [6]. Since omega-3 fatty acids are highly unsaturated because of the presence of multiple double bonds, they are highly susceptible to oxidation and thermal degradation, which may lead to the formation of free radicals and hydroperoxides [7]. Lipid hydroperoxides are the main products of lipid oxidation, and they decompose rapidly to form ketones, aldehydes, alcohols, hydrocarbons, volatile organic compounds and epoxy compounds [8]. Although many studies have been conducted on the oxidation of seafood lipids immediately after frying, more is needed to know about their lipid oxidation when stored in the refrigerator.

Most vegetable oils used for the deep-fat frying of seafood contain some natural antioxidants, such as tocopherols [9]. However, because these primary antioxidants are mostly present in small quantities, their ability to stabilize oils under extreme conditions is limited. Additionally, some tocopherols can act as pro-oxidants at specific concentrations [10]. Therefore, additional antioxidants may be required when vegetable oils are used for frying to prevent the degradation of fried foods.

Antioxidants are a group of compounds used to prevent the production of free radicals in lipids. They work by interfering with the free-radical formation mechanism, thereby slowing down lipid oxidation. The most widely used ones include butylated hydroxytoluene (BHT), butylated hydroxyanisole (BHA), propyl gallate (PG) and Tert-butylhydroquinone (TBHQ). While these are potent antioxidants, they are limited in their use at elevated temperatures. For instance, PG, TBHQ, BHA and BHT decompose and volatilize at temperatures lower than 110 °C [11], making them unsuitable for use during the deep frying of food at 170 °C. Also, they are chemically synthesized and have been reported to have some cytotoxic and apoptotic activities [12]. Thus, attention is shifting towards natural, food-derived antioxidants because they are safe and are mostly thermostable. Also, most of them are abundant in agri-food waste and some food industry by-products. For instance, hydroxytyrosol—a polyphenol derived from the olive plant—is more abundant in the by-products of olive oil production, such as olive mill wastewater and pomace, than in olive oil [13]. Also, hydroxytyrosol is one of the few polyphenols that is soluble in oil and water and is considered the most potent antioxidant after gallic acid [14]. These unique properties have led to an increase in interest in its recovery and exploitation.

Therefore, in this study, the inhibitory effects of hydroxytyrosol, α-tocopherol and ascorbyl palmitate on lipid peroxidation in squid, prawn and hoki fish during deep-fat frying and refrigerated storage were studied. Hydroxytyrosol, α-tocopherol and ascorbyl palmitate were selected because they are lipophilic and thermostable [11,15,16]. We studied the changes in the lipid profiles of the seafood. We used several techniques to determine the ability of the antioxidants to slow down the formation of primary and secondary oxidation products in deep-fried seafood.

## 2. Materials and Methods

### 2.1. Materials

Fresh Hoki fish (Blue grenadier), squids (Loligo squid) and prawns (Tiger) were purchased from the local seafood shop in New South Wales (NSW), Australia. They were cleaned, eviscerated and washed with distilled water before use. The frying medium (sunflower oil), with no added antioxidants, was purchased from a local supermarket. The main fatty acids present in the sunflower oil (SFO) were determined using GC to be palmitic acid (C16:0) 6.2%, stearic acid (C18:0) 3.5%, oleic acid (C18:1) 21.7% and linoleic acid (C18:2) 66.5%. Ascorbyl palmitate, α-tocopherol, gas chromatography standards and other chemicals, including toluene, butylated hydroxytoluene (BHT), acetyl chloride, methanol, chloroform, sodium chloride, potassium bicarbonate, *p*-Anisidine, 2-Thiobarbituric acid and 2,2-Diphenyl-1-picrylhydrazyl (DPPH), were purchased from Sigma Aldrich (Castle Hill, NSW, Australia). Hydroxytyrosol was a gift from Professor Colin Barrow, Deakin University, Geelong, Australia.

### 2.2. Proximate Composition

The moisture, ash and crude protein contents of the seafood were determined according to the Association of Official Analytical Chemists standard methods [1]. Briefly, the moisture content was determined by drying each sample at 105 °C to a constant weight (achieved within 8 h). The ash content was determined by the combustion of a 5 g sample at 525 °C for 5 h, while the crude protein content was determined by using the Kjeldahl procedure with a 2 g sample.

### 2.3. Lipid Extraction

Lipids were extracted following a previously reported method [2] with some modifications. Briefly, 50 g of homogenised hoki fillet, squid and prawn were placed in a borosilicate glass bottle containing 200 mL of chloroform-methanol mix (1:1). The bottle was screw-capped and left for 1 h at room temperature under stirring. Next, 50 mL of deionised water was added to the mixture, thoroughly mixed and centrifuged at 4000 rpm for 10 min. The organic solvent was collected and removed by rotary evaporation, and the resulting lipid was weighed, kept at −18 °C and analysed within 48 h.

### 2.4. Analysis of Fatty Acid Composition by Gas Chromatography

The fatty acids in the seafood lipids were converted to methyl esters, and the resulting fatty acid methyl esters (FAMES) were analysed by an Agilent 6890 gas chromatograph with a flame ionisation detector (FID), as previously described [3].

### 2.5. Concentration and Thermal Stability of Antioxidants

In order to determine the appropriate amount of antioxidants needed for deep frying, three concentrations (0.5, 1.0 and 1.5 mmol/kg oil) of each antioxidant were added to the sunflower oil and fried at 170 °C for 10 min without sample frying. A 3 L electrical deep fat fryer (Anko, TP-1302, 2100 W, Mulgrave, Australia) equipped with a thermostat, timer and stainless-steel wire basket was used for the frying experiment. The oil was allowed to cool to room temperature and tested for the presence of hydroperoxides, as explained in Section 2.8.1.

### 2.6. Thermal Stability of Antioxidants

The ability of each antioxidant to withstand the frying temperature (170 °C) for longer was investigated by adding the predetermined concentration to the SFO. The oil was then fried at 170 °C for 120 min without sample frying. Oil samples were collected every 20 min, cooled to room temperature and tested for the presence of hydroperoxides, as explained in Section 2.8.1.

### 2.7. Frying Process

The cleaned squid, prawn and hoki fillets were divided into eight groups for frying in the sunflower oil (two for control groups—days 0 and 2), and two of each were fried in sunflower oils containing predetermined concentrations of α-tocopherol (days 0 and 2), hydroxytyrosol (days 0 and 2) and ascorbyl palmitate (days 0 and 2). The sunflower oils were heated to 170 °C, and the seafood placed in a frying basket was lowered for deep frying for 10 min (seafood/sunflower oil, 40:100, *w*/*v*). After frying, the seafood in the basket was emptied into a stainless-steel filter net for 25 min to drain the sunflower oil and was then pat-dried using kitchen paper. The fried seafood was divided into two equal portions. The lipids from one portion were extracted and analysed immediately, and these were designated “day 0” samples. Meanwhile, the second portion was kept inside silicone food storage bags with an airtight seal and stored in the refrigerator at 0 °C for two days before lipid extraction and analysis, and they were designated “day 2”. We chose to store the fried seafood in the fridge for a maximum of two days because this is generally the longest time that fried squids can be kept under this condition before they start losing their sensory properties [4].

### 2.8. Oxidation Stability Test

#### 2.8.1. Peroxide Value (POV)

A POV test was carried out following the American Oil Chemists’ Society (AOCS) method Cd 8b-90 [5] with some modifications. Briefly, a 0.2 g lipid sample was dissolved in 50 mL of iso-octane and glacial acetic acid mix (2:3 *v*/*v*), and the solution was swirled to ensure the complete dissolution of the contents. After the addition of 0.5 mL of saturated potassium iodide, the solution was mixed for 60 s, followed by an addition of 30 mL of distilled water. The solution was titrated with 0.05 N sodium thiosulfate until the yellow iodine colour had almost disappeared, and then 0.5 mL of 10% sodium lauryl sulfate and 0.5 mL of a starch indicator solution were added. The titration was continued until the blue colour disappeared. A blank titration was carried out using the reagent without the sample, and the peroxide value was calculated using the formula POV = [0.05(*S* − *B*)1000]/*m.* Where *S* and *B* are the volume (mL) of the titrant for the sample and the blank, respectively, and *m* is the sample mass (g). The POV was expressed as milli-equivalents of peroxide per kilogram of lipid (meq/kg lipid).

#### 2.8.2. Thiobarbituric Acid Reactive Substance (TBARS)

The TBARS tests were carried out as described previously with some modifications [6]. Briefly, 100 mg of lipid sample was mixed with 25 mL of 1-butanol. Then, 5 mL of the mixture was transferred into a screw-capped bottle containing 5 mL of freshly prepared TBA reagent (0.5 g of TBA in 250 mL of 1-butanol) and thoroughly mixed. Contents were incubated at 95 °C for 2 h, cooled on ice and centrifuged at 6500 rpm for 5 min. The clear-coloured liquid was transferred to a clean cuvette, and absorbance was read at 535 nm in a Cary 60 UV-Visible Spectrophotometer (Agilent Technologies; Santa Clara, CA, USA). The results were calculated using the formula *C* = (0.415 × *A*)/*w*, where *A* is the absorbance at 532 nm, *w* is the sample mass (g), and factor 0.415 was obtained from the calibration curve of the MDA standard (1,1,3,3-tetramethoxypropane). The final results of TBARS were expressed as milligram (mg) MDA equivalents per kilogram (kg) of lipid sample.

#### 2.8.3. *p*-Anisidine Value (*p*-AV)

The *p*-Anisidine test was carried out according to a previously described method [7] using 0.25 g of lipid sample. The results were calculated using the formula *p-*AV = [25(1.2*A*_2_ − *A*_1_)]/*m,* where *A_1_* and *A_2_* are the absorbance values before and after adding *p-*anisidine; and *m* is the sample mass (g).

### 2.9. Acid Value (AV)

The AV was determined according to the American Oil Chemists’ Society (AOCS) official method Cd 3d-63-90 [5] with some modifications. Briefly, the lipid sample (0.2 g) was dissolved in 50 mL of isopropanol and diethyl–ether mix (1:1 *v*/*v*) in a 100 mL bottle. Contents were mixed and titrated with 0.01 M potassium hydroxide (KOH). The AV was calculated using the formula AV = [(*A* − *B*) × *M* × 56.1]/*W*, where *A* is the volume of KOH used in the titration (mL), *B* is the volume of KOH used in titrating the blank (mL), *M* is the concentration of KOH (mol/L), and *W* is the mass of the oil sample (g). The final results (AV) were expressed as milligrams of KOH per gram of lipid sample (mg KOH/g lipid).

### 2.10. Statistical Analysis

Unless stated otherwise, the experiments were carried out in triplicates, and the results are presented as the mean ± standard deviation (SD). Statistical significance was evaluated using one-way analysis of variance (ANOVA), and multiple comparisons were achieved by Tukey-Kramer HSD (honestly significant difference). The mean differences were significant when the probability was less than 0.05 (*p* < 0.05). Statistical analyses were performed using SPSS 26.0 (SPSS Inc., Chicago, IL, USA).

## 3. Results and Discussion

### 3.1. Lipid Content of Fresh Seafood

The total lipid content analyses of the seafood were carried out, and the results showed that hoki has the highest amount of lipid (1.9 g/100 g wet weight), followed by squid (1.3 g/100 g wet weight) and prawn (1.1 g/100 g wet weight). While these values are low, they are consistent with previous reports that most Australian seafood species are generally low in fat [8,9]. The average lipid content of Australian fish range between 0.5 g and 2.7 g/100 g (wet basis), covering commonly consumed seafood, including hoki (blue grenadier), squid and prawn [9]. Interestingly, the lipid content of the hoki used in this current study is higher (1.9 g/100 g) than those reported for rainbow trout (1.7 g/100 g), barramundi (1.2 g/100 g), Australian herring (1.2 g/100 g), Australian salmon (1.0 g/100 g) and southern bluefin tuna (0.7 g/100 g) [9].

### 3.2. Fatty Acid Composition by Gas Chromatography

The fatty acid compositions of hoki, squid and prawn oils are presented in Table 1. The major saturated fatty acids (SFAs) in the oils include myristic (C14:0), palmitic (C16:0), Stearic (C18:0) and arachidic (C20:0) acids. As shown (Table 1), squid oil has significantly higher (*p* < 0.05) total SFA (40.43 ± 0.40%) than prawn (30.32 ± 0.54%) and hoki (29.60 ± 0.33%) oils. These results are similar to those previously reported by Soltan and Gibson (2008). Meanwhile, the total MUFAs in hoki oil, which include oleic acid (C18:1n9), palmitoleic acid (C16:1n9) and vaccenic acid (C18:1n7), is higher (27.40 ± 0.41%) than in prawn (22.82 ± 0.55%) and squid (6.35 ± 0.30%) oils.

The polyunsaturated fatty acid (PUFA) results show that all three seafood samples are rich sources of omega-3 PUFAs (Table 1). Prawn oil has equal levels of EPA (15.11 ± 0.27%) and DHA (15.84 ± 0.24%), while squid and hoki oils have more DHA than EPA (Table 1). The total n-3 fatty acids in the oils are 46.03 ± 0.55% (squid), 36.40 ± 0.44% (hoki) and 33.00 ± 0.43% (prawn). The levels of EPA and DHA in hoki oil are similar to those reported for tuna and bonito oils (EPA-5%; DHA-25%) [10,11], while the levels in prawn oil are similar to those in anchovy oil (EPA-17%; DHA-12%) (Akanbi et al., 2013). These results show that all the seafood tested in this study are high in omega-3 fatty acids. Meanwhile, arachidonic acid (C20:4n6)—the major omega-6 (*n*-6) PUFA in the seafood samples—is significantly (*p* < 0.05) higher in prawn oil (8.19 ± 0.13%) than in squid (3.70 ± 0.00%) and hoki (2.90 ± 0.16%) oils. Lipids containing high levels of n-3 PUFAs and lower levels of *n*-6 PUFAs (as seen in hoki, squid and prawn) are healthy and are recommended for reducing the risk of many chronic diseases, including cardiovascular, inflammatory and autoimmune diseases [12,13].

### 3.3. Concentration and Stability of Antioxidants

Although the native tocopherols in most vegetable oils may be sufficient to protect them against deterioration at room temperature, more efficient antioxidants at adequate concentrations are needed when the oils are subjected to extreme temperature treatments like deep-fat frying. The effects of different antioxidant concentrations on inhibiting the oxidation of sunflower oil (SFO) during frying (without the seafood) were investigated by measuring its peroxide value (POV). The POV is an indicator of primary oxidation products during the early stages of lipid oxidation, and so was used for determining the effectiveness of the antioxidants. The results in Figure 1 shows that the POV increased significantly in the SFO without antioxidants. Meanwhile, all the antioxidants reduced the POV in the fried oil when used at a concentration of 0.5 mmol/kg, but hydroxytyrosol was the most effective. However, at 1.0 and 1.5 mmol/kg concentrations, all the antioxidants were highly effective (Figure 1). Therefore, a 1.0 mmol/kg antioxidant concentration was selected as the most suitable for deep-fat frying in this study.

Furthermore, using a 1.0 mmol/kg concentration, we investigated the stability of the antioxidants in SFO over an extended frying period since studies have shown that antioxidants in oils do undergo thermal degradation during frying [14,15]. As shown (Figure 2), the POV formation increased significantly when no antioxidant was added to the SFO, however, a significant inhibition of lipid peroxidation was observed in the presence of antioxidants with slight increases in the POV. Although a slightly higher POV was recorded for α-tocopherol compared to ascorbyl palmitate and hydroxytyrosol, they all still showed high thermal stability and antioxidant activity. These results are not surprising because previous thermogravimetric analyses of the antioxidants showed that α-tocopherol, ascorbyl palmitate and hydroxytyrosol had thermal decomposition temperatures of 199, 240 and 262 °C, respectively [16,17,18], which are higher than the 170 °C used for frying in this study. We have established that the antioxidants are thermostable in SFO without frying seafood. However, when the oil is used for frying seafood, the complex interaction involving water, oxygen, frying oil and the seafood components may overwhelm the antioxidants and alter their activity significantly. So, these have to be studied further.

### 3.4. Changes in the Proximate Composition and Lipid Profiles of Seafood

The changes in the proximate composition of hoki, squid and prawn before and after deep-fat frying were determined and reported on a wet weight basis. The results presented in Table 2 were obtained from the control group (without antioxidants) since treatment with antioxidants did not affect the proximate composition of the seafood. As shown (Table 2), the moisture contents of the fresh seafood were 82.30 ± 1.92 g/100 g (squid), 80.52 ± 1.52 g/100 g (hoki) and 73.32 ± 1.02 g/100 g (prawn). After frying for 10 min, their moisture contents had been reduced by about 60% (squid), 70% (hoki) and 75% (prawn), respectively, due to heat-induced evaporation [19].

Meanwhile, the protein, lipid and ash contents of the fried seafood increased significantly (*p* < 0.05) because of water displacement during frying [20]. For instance, the protein content of prawn had increased from 20.10 to 36.84 g/100 g, followed by squid’s (13.84 to 31.92 g/100 g) and hoki’s (12.13 to 29.63 g/100 g). Also, due to the absorption of the SFO used for frying, the lipid contents of the seafood increased significantly, with hoki having the highest (31.20 g/100 g), followed by prawn (30.61 g/100 g) and squid (19.32 g/100 g). The low lipid absorption by squid during frying was due to its firm and bouncy texture, unlike hoki and prawn. Meanwhile, for the ash contents, the highest increase was observed in squid, followed by hoki and prawn (Table 2). Consistently, previous studies reported decreases in moisture contents and increases in protein, fat and ash contents of a broad range of deep-fried seafood [20,21,22].

### 3.5. Changes in the Peroxide Values (POV) of the Fried Seafood

The results presented in Figure 3 show the POVs of the oils extracted from the seafood after frying (day 0) and refrigerated storage (day 2). The POV values of the control (with no antioxidants) were significantly higher (*p* < 0.05) than those with antioxidants. Also, although squid had the highest levels of highly oxidisable n-3 PUFAs (Table 1), its POV value in the control sample on day 0 (Figure 3a) was lower (6.15 ± 0.21 meq/kg) than the 8.68 ± 0.42 meq/kg recorded for hoki (Figure 3b) and 8.32 ± 0.21 meq/kg for prawn (Figure 3c). It appeared that the POV values followed the trend of the levels of absorbed SFO by each seafood during frying (shown in Table 2). These indicate that lipid oxidation occurred more in the SFO than in the seafood during frying. However, after 2 days of refrigerated storage, the POV increased significantly (*p* < 0.05) in squid (Figure 3b), despite absorbing less SFO than hoki and prawn. These show that the oxidation of n-3 PUFAs in the fried squid continued despite the refrigerated storage. These results are consistent with previous studies that the low-temperature storage of seafood does not slow down or stop lipid peroxidation [23,24].

Meanwhile, the antioxidants inhibited lipid oxidation in the SFO and seafood, albeit differently. The most effective antioxidant during storage was hydroxytyrosol followed by ascorbyl palmitate, while α-tocopherol was the least effective. Results show that with α-tocopherol, rapid lipid peroxidation occurred. A higher POV value was seen in squid (10.72 ± 0.40 meq/kg), followed by hoki (9.82 ± 0.42 meq/kg) and prawn (8.40 ± 0.42 meq/kg) after two days of refrigerated storage (Figure 3a–c). Although ascorbyl palmitate was better than α-tocopherol, hydroxytyrosol was the most effective, with very slight increases in the POV (less than 1.0 meq/kg) from day 0 to day 2 for all the seafood.

While these antioxidants have not been previously compared under frying conditions, studies have shown that hydroxytyrosol is a more potent antioxidant than α-tocopherol. For instance, a study found that the time taken for stripped olive oil to reach a peroxide value of 50 Meq/kg at 60 °C in the presence of hydroxytyrosol was 70 days, meanwhile with α-tocopherol, it took 20 days [25]. Also, when stripped olive oil samples containing 1 mmol/kg antioxidant were subjected to accelerated oxidation in a Rancimat apparatus at 100 °C, the oil containing hydroxytyrosol had an induction time (an oil stability index) of 100 h, while that containing α-tocopherol had an induction time of 50 h [26], meaning that hydroxytyrosol can keep the oil fresher for longer than α-tocopherol. An older study using the Rancimat apparatus at 120 °C also found hydroxytyrosol to be more effective than α-tocopherol in stabilizing olive oil [27].

The effectiveness of ascorbyl palmitate versus α-tocopherol could be due to its surface activity [28,29]. Surface active antioxidants are mostly responsible for inhibiting the oxidation of lipids at the oil-water interface [30]. Since the seafood contained an appreciable amount of water, the water released during frying, which is essentially insoluble in oil, could be in the form of association colloids, thereby producing lipid hydroperoxides that are surface active [31]. Therefore, surface active antioxidants like ascorbyl palmitate could inhibit the production of these hydroperoxides. A study found that the surface activity of ascorbyl palmitate was responsible for the enhanced oxidative stability of flaxseed oil at 110 °C. However, α-tocopherol was highly ineffective at this temperature [28].

### 3.6. Changes in the p-Anisidine Values (p-AV) of the Fried Seafood

The flavour and odour of oils deteriorate during frying as a result of the accumulation of aldehydes and ketones that are produced when hydroperoxides decompose [32]. The results presented in Figure 4 show significant increases (*p* < 0.05) in the *p*-AV from day 0 to day 2 in the control samples. Again, the higher *p*-AV values seen on day 0 were due to increased oxidation of SFO absorbed by the seafood during frying.

Meanwhile, after the refrigerated storage of the fried seafood (day 2), higher *p*-AV was observed in the lipids extracted from squid than in hoki and prawn. This was because squid had higher levels of n-3 PUFAs, which had undergone rapid oxidation. Previous research also found a positive correlation between the n-3 PUFA contents and aldehyde formation in fried seafood [33,34]. These increases of *p*-AV in the seafood lipids further suggest that lipid hydroperoxides are decomposed into secondary oxidation products under the deep-frying process.

As expected, the antioxidants had different inhibitory effects. As shown in Figure 4, hydroxytyrosol inhibited the formation of the secondary oxidation products more efficiently than ascorbyl palmitate and α-tocopherol in all the seafood. With hydroxytyrosol, slight increases in *p*-AV were recorded after refrigerated storage for 2 days. However, with ascorbyl palmitate and α-tocopherol, the *p*-AV obtained was doubled after 2 days, although ascorbyl palmitate was a better antioxidant of the two. A study found that hydroxytyrosol was about three times more effective than α-tocopherol in suppressing aldehyde (*p*-AV) formation in olive oil triacylglycerols at 60 °C [25]. Therefore, a more potent antioxidant like hydroxytyrosol could inhibit the accumulation of aldehydes and ketones during the deep-fat frying of PUFA-rich foods.

### 3.7. Changes in the TBARS Values of the Fried Seafood

The thiobarbituric acid reactive substances test (TBARS) is used to determine the presence of malondialdehyde (MDA), a secondary oxidation product formed when hydroperoxides decompose. The MDA reacts with thiobarbituric acid (TBA) to form a pink compound (TBARS), which is measured at 535 nm [6,35].

The results presented in Figure 5 show that the antioxidants were able to suppress MDA formation in the seafood during frying (day 0) compared to the control. As shown, the most effective antioxidant was hydroxytyrosol, while ascorbyl palmitate performed slightly better than α-tocopherol in all the seafood. After two days of refrigerated storage, the TBARS values in the control had doubled the day 0 values, indicating rapid oxidation of omega-3 lipids in the seafood and the absorbed SFO. Similar doubling trends were found with the α-tocopherol and ascorbyl palmitate-treated samples, although they had lower TBARS values than the control samples. A previous study found an increase in MDA formation in fish crackers fried at 170 °C for 1 min in a tocopherol-rich soybean oil. The authors attributed this to the degradation of tocopherol during deep-fat frying [36]. Meanwhile, with hydroxytyrosol, a small increase in TBARS (<0.35 mg MDA/kg lipid) was observed in each seafood from day 0 to day 2, showing the effectiveness of this antioxidant in suppressing MDA formation in fried seafood during refrigerated storage. These results agree with previous studies showing that hydroxytyrosol could inhibit MDA formation in different PUFA-enriched meat products during chilling storage [37,38].

### 3.8. How Effective Are the Antioxidants in SFO for Multiple Frying Cycles?

In order to determine the number of frying cycles that can be performed using antioxidant-treated SFO, fresh squids were fried multiple times at 170 °C. A total of 7 frying cycles were carried out in this study. After each frying cycle (10 min), the fryer was turned off, and the SFO was allowed to cool down to room temperature. The SFO samples were collected after each frying cycle, and their acid value (AV) and POV were analysed, with results presented in Figure 6. As expected, rapid oxidation occurred in the control sample because of the lack of antioxidants. The water migration from each batch of the squid to the SFO increased its hydrolysis, leading to free fatty acid (FFA) formation, as reflected by the AV content of the oil (Figure 6b). Meanwhile, some interesting results were seen in the antioxidant-treated SFO. As shown (Figure 6a), hydroxytyrosol was more effective than α-tocopherol and ascorbyl palmitate in suppressing POV for the first 3 frying cycles. However, its POV rose sharply afterwards, making it less effective than ascorbyl palmitate but better than α-tocopherol. Similar trends were observed in the AV results (Figure 6b). Both the POV and AV results showed that the ascorbyl palmitate remained active throughout the frying period with slow and steady increases. Therefore, ascorbyl palmitate was a better antioxidant for multiple frying of squid.

The short frying cycles recorded for hydroxytyrosol and α-tocopherol in this study may not be entirely due to thermal degradation because they are reasonably thermostable, although hydroxytyrosol was better (Figure 2). Two possible explanations for these observations are that the antioxidants could have been rapidly used up during frying or absorbed by the seafood. A study found that the decrease in the concentrations of hydroxytyrosol and tyrosol in virgin olive oil used for frying fish was due to a significant enrichment of these compounds in the fried fish and not due to thermal degradation [39]. Similar observations were reported in another study, but the authors also found that polyphenols are more enriched in foods than tocopherols during frying because they are more polar [40]. This could explain why hydroxytyrosol was more enriched in fried Mediterranean finfish than α-tocopherol [39]. So, the ineffectiveness of α-tocopherol in this study could be because it was rapidly used up during the multiple frying and not due to significant absorption in the fried squid. It has been reported that tocopherols in frying oils are decomposed by both direct oxidation with oxygen and reaction with oxidized fatty acids [41].

Meanwhile, it appeared that hydroxytyrosol was highly enriched in the fried squid, and this enrichment meant greater protection of lipids against oxidation during frying and storage. This could explain why the lipids extracted from the fried squid, hoki and prawn in the presence of hydroxytyrosol were more stable after frying and during refrigerated storage compared to when α-tocopherol and ascorbyl palmitate were used as antioxidants (Figure 3, Figure 4 and Figure 5). In any event, although hydroxytyrosol is not suitable for multiple frying cycles like ascorbyl palmitate, it remains a potent antioxidant for the deep-fat frying of seafood.

## 4. Conclusions

This present study demonstrated that the deep-fat frying and refrigerated storage of hoki, prawn and squid resulted in time-dependent oxidation changes in their lipids and the frying medium, which is sunflower oil. Lipid oxidation tests, including POV, *p*-AV and TBARS, showed that oxidation occurred during the frying and refrigerated storage of the seafood. The presence of antioxidants, including hydroxytyrosol, α-tocopherol and ascorbyl palmitate slowed lipid oxidation, but hydroxytyrosol was the most effective. It effectively reduced the oxidation of PUFA and the generation of aldehydes in the fried seafood and SFO. However, with hydroxytyrosol, multiple frying cycles could not be achieved as it appeared that the antioxidant was absorbed in the seafood during frying, thus offering additional protection for the PUFAs within the seafood. Meanwhile, ascorbyl palmitate remained active throughout the multiple frying periods with slow increases in POV and AV, unlike α-tocopherol, which appeared to have been rapidly used up or decomposed by direct oxidation with oxygen and reaction with oxidized fatty acids.

These results suggest that hydroxytyrosol is a good antioxidant for use in SFO during the deep-fat frying of seafood, and it would prevent the oxidation of the PUFAs in the seafood during refrigerated storage. For consumers, these findings show that oils rich in polyphenols such as hydroxytyrosol could be used for deep-fat frying of seafood but for limited reuses, which could be a healthy way to process and consume seafood.

Since olive-derived polyphenols, such as hydroxytyrosol and tyrosol, are abundant in olive processing waste, and several studies have shown that these compounds can be cost-effectively extracted, future studies should focus on their food utilization. For instance, cheaper seed oils with added natural antioxidants such as hydroxytyrosol should be investigated further because these may be an economical way of industrial production of polyphenol-rich oils that may be cheaper than olive oil, making it available to many consumers.

This study demonstrated that natural antioxidants could be used instead of chemically synthesized ones for suppressing lipid peroxidation during deep-fat frying and refrigerated seafood storage.

## Figures and Tables

**Figure 1 antioxidants-12-00929-f001:**
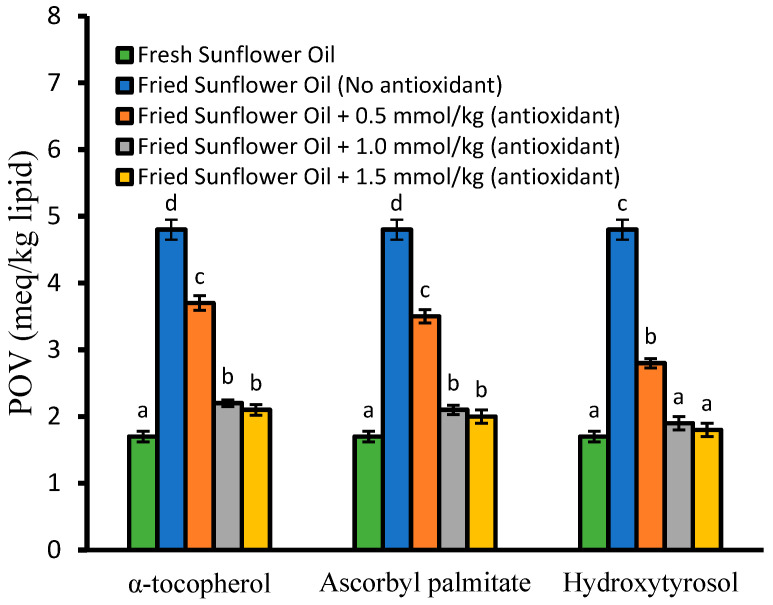
The effect of antioxidant concentration on the stability of sunflower oil (SFO). Values are expressed as mean ± standard deviation (*n* = 3). Bars not sharing a common letter are significantly different (*p* < 0.05) within each treatment.

**Figure 2 antioxidants-12-00929-f002:**
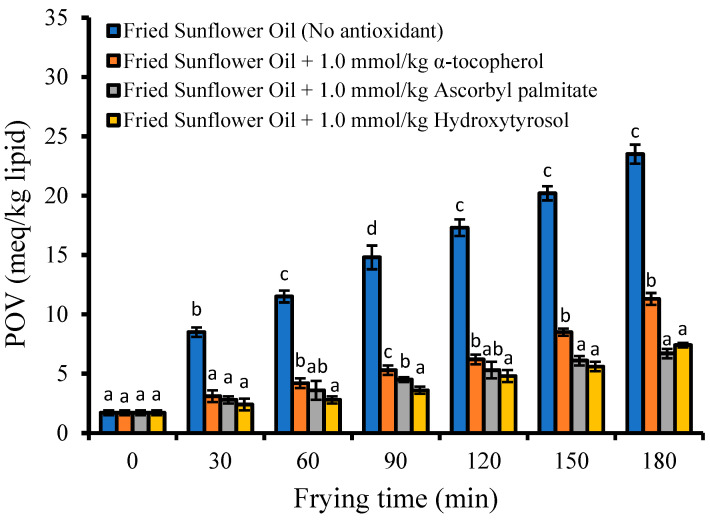
The stability of antioxidants in sunflower oil (SFO) over an extended frying period. Values are expressed as mean ± standard deviation (*n* = 3). Bars not sharing a common letter are significantly different (*p* < 0.05) within each treatment.

**Figure 3 antioxidants-12-00929-f003:**
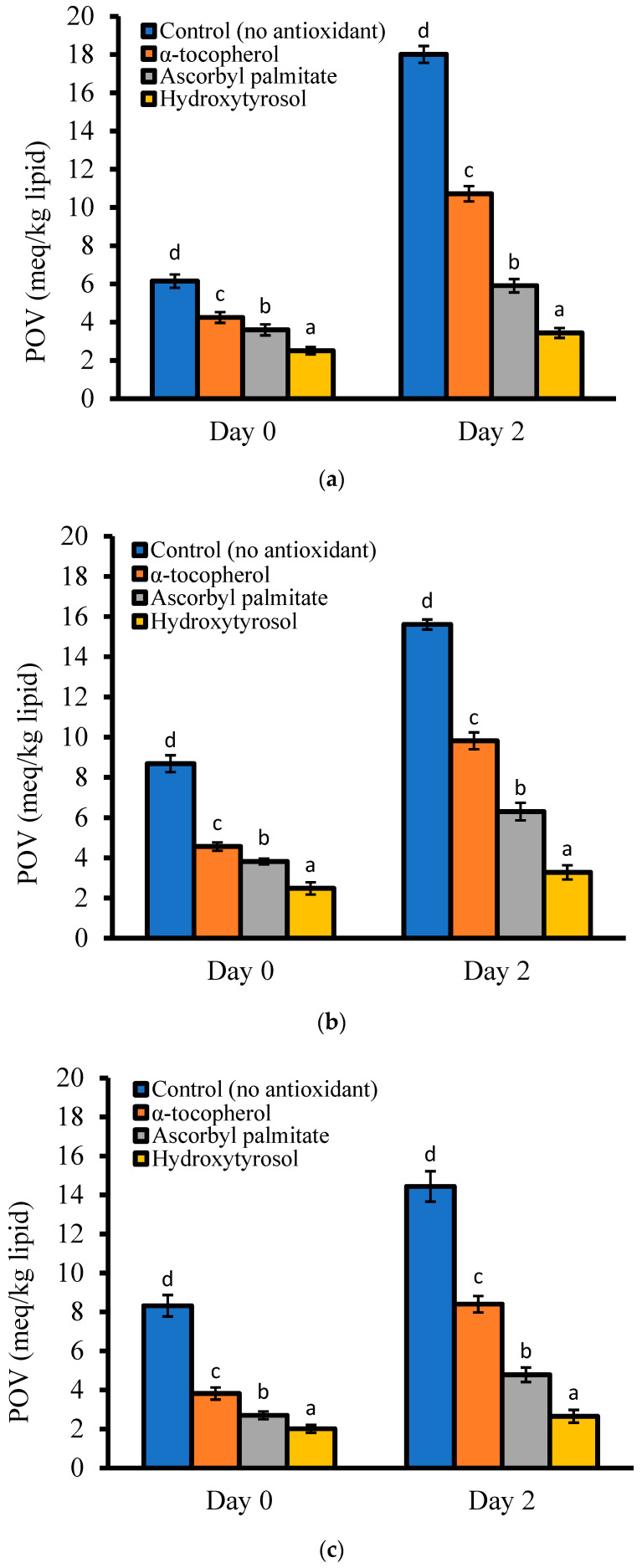
The peroxide values (POV) (meq/kg) of oil extracted from (**a**) squid, (**b**) hoki and (**c**) prawn after deep frying and refrigerated storage. Values are expressed as mean ± standard deviation (*n* = 3). Bars not sharing a common letter are significantly different (*p* < 0.05) within each treatment.

**Figure 4 antioxidants-12-00929-f004:**
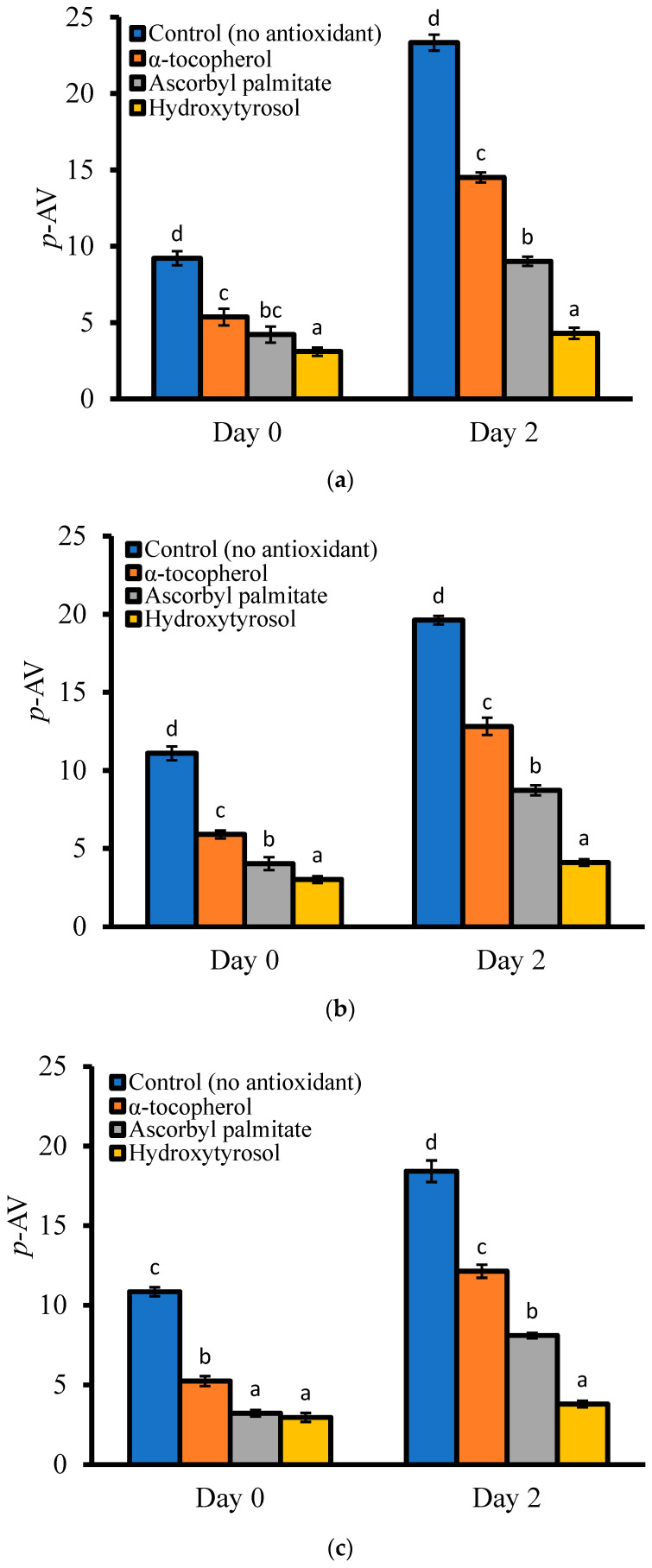
The *p*-Anisidine values (*p*-AV) of oil extracted from (**a**) squid, (**b**) hoki and (**c**) prawn after deep frying and refrigerated storage. Values are expressed as mean ± standard deviation (*n* = 3). Bars not sharing a common letter are significantly different (*p* < 0.05) within each treatment.

**Figure 5 antioxidants-12-00929-f005:**
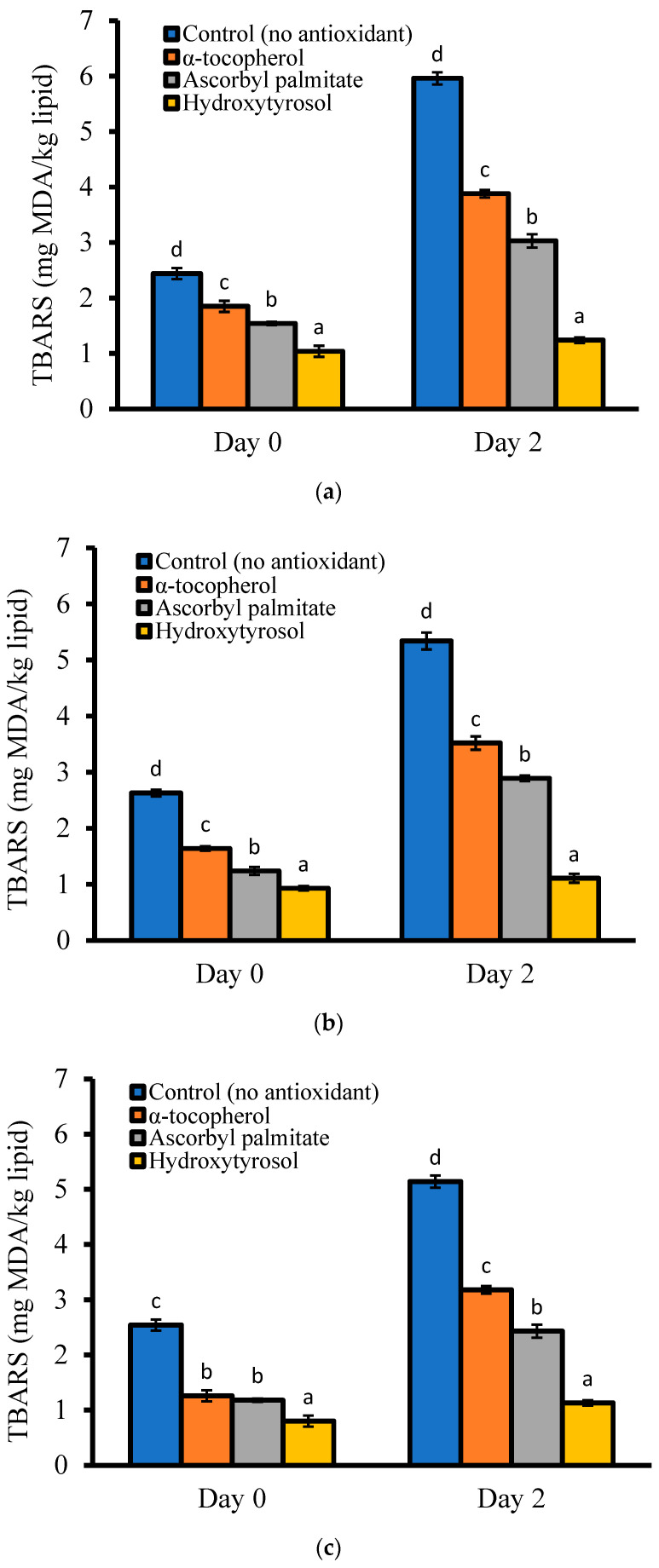
The thiobarbituric acid reactive substances (TBARS) values of oil extracted from (**a**) squid, (**b**) hoki and (**c**) prawn after deep frying and refrigerated storage. Values are expressed as mean ± standard deviation (*n* = 3). Bars not sharing a common letter are significantly different (*p* < 0.05) within each treatment.

**Figure 6 antioxidants-12-00929-f006:**
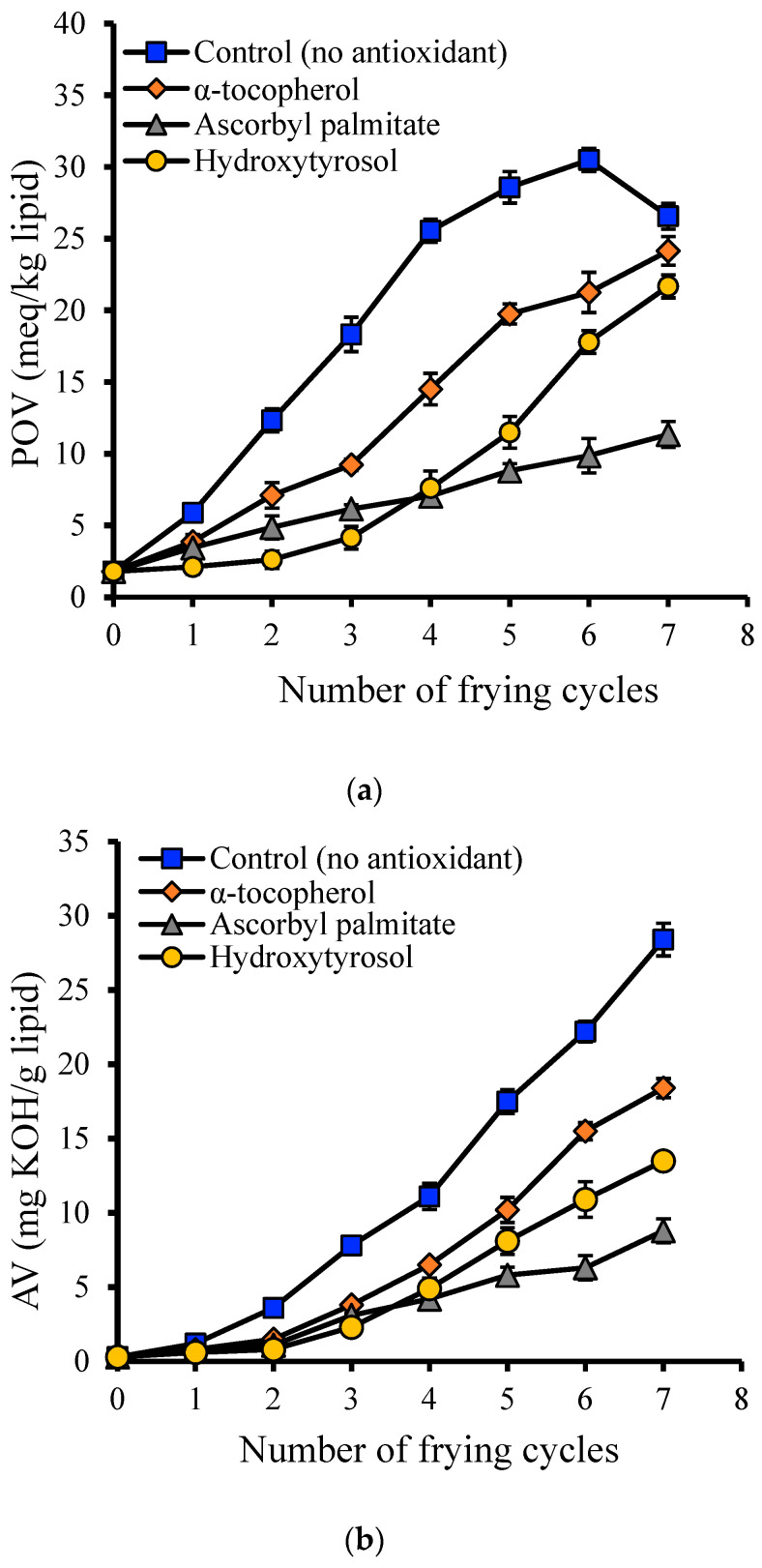
Changes in the (**a**) POV (meq/kg lipid) and (**b**) AV (mg KOH/g lipid) of SFO during repeated deep frying of squid. Results are mean ± SD for triplicate replications.

**Table 1 antioxidants-12-00929-t001:** Fatty acid compositions of lipids from hoki, squid and prawn.

Fatty Acid (%)	Squid	Hoki	Prawn
C14:0	2.60 ± 0.01 ^b^	3.10 ± 0.05 ^c^	1.21 ± 0.00 ^a^
C15:0	0.31 ± 0.01 ^a^	1.00 ± 0.10 ^b^	1.72 ± 0.01 ^c^
C16:0	29.51 ± 0.10 ^c^	7.68 ± 0.11 ^b^	13.27 ± 0.21 ^a^
C17:0	0.80 ± 0.02 ^a^	0.92 ± 0.03 ^b^	2.31 ± 0.02 ^b^
C18:0	4.91 ± 0.01 ^b^	3.01 ± 0.01 ^a^	11.24 ± 0.17 ^c^
C20:0	2.30 ± 0.05 ^b^	3.89 ± 0.02 ^c^	0.57 ± 0.15 ^a^
Total SFA	40.43 ± 0.40 ^b^	29.60 ± 0.33 ^a^	30.32 ± 0.54 ^a^
C16:1n9	0.40 ± 0.01 ^c^	0.10 ± 0.00 ^a^	0.31 ± 0.02 ^b^
C16:1n7	0.51 ± 0.05 ^a^	4.89 ± 0.11 ^b^	5.73 ± 0.22 ^c^
C17:1n9	0.61 ± 0.05 ^ab^	0.41 ± 0.02 ^a^	0.46 ± 0.14 ^a^
C18:1n9 OA	2.80 ± 0.02 ^a^	8.01 ± 0.13 ^c^	13.10 ± 0.19 ^b^
C18:1n7	0.92 ± 0.01 ^a^	2.78 ± 0.10 ^c^	1.10 ± 0.07 ^b^
C20:1n9	0.50 ± 0.05 ^a^	0.41 ± 0.05 ^a^	0.70 ± 0.06 ^b^
C22:1n9	0.61 ± 0.05 ^a^	0.80 ± 0.11 ^b^	1.41 ± 0.02 ^c^
Total MUFA	6.35 ± 0.30 ^a^	27.40 ± 0.41 ^c^	22.82 ± 0.55 ^b^
C18:3n3 ALA	0.30 ± 0.01 ^a^	0.28 ± 0.01 ^a^	0.73 ± 0.02 ^b^
C20:5n3 EPA	13.21 ± 0.12 ^b^	4.91 ± 0.16 ^a^	15.11 ± 0.27 ^c^
C22:5n3 DPA	1.01 ± 0.03 ^a^	2.31 ± 0.10 ^c^	1.32 ± 0.03 ^b^
C22:6n3 DHA	31.51 ± 0.33 ^c^	28.90 ± 0.34 ^b^	15.84 ± 0.24 ^a^
Total n-3 PUFA	46.03 ± 0.55 ^c^	36.40 ± 0.44 ^b^	33.00 ± 0.43 ^a^
C18:2n6 LA	0.20 ± 0.00 ^a^	0.50 ± 0.03 ^b^	0.61 ± 0.01 ^c^
C20:4n6 AA	3.50 ± 0.03 ^b^	2.40 ± 0.15 ^a^	7.58 ± 0.12 ^c^
Total n-6	3.70 ± 0.03 ^b^	2.90 ± 0.16 ^a^	8.19 ± 0.13 ^c^
Others *	3.60 ± 0.10 ^a^	3.70 ± 0.02 ^a^	5.67 ± 0.04 ^b^

Values are mean ± SD for duplicate replications. Mean values in the same row not sharing a common superscript are significantly different (*p* < 0.05). SFA, saturated fatty acids; MUFA, monounsaturated fatty acids; PUFA, polyunsaturated fatty acids; OA, oleic acid; ALA, α-linoleic acid; EPA, eicosapentaenoic acid; DPA, docosapentaenoic acid; DHA, docosahexaenoic acid; LA, linolenic acid; AA, arachidonic acid. * Others—sum of unknown fatty acid peaks.

**Table 2 antioxidants-12-00929-t002:** Proximate composition of seafood.

Sample	Treatment	(g/100 g, Wet Weight Basis)			
		Moisture	Protein	Lipid	Ash
Squid	Fresh	82.30 ± 1.92 ^b^	13.84 ± 0.93 ^a^	1.30 ± 0.11 ^a^	1.12 ± 0.02 ^b^
	Deep-fried *	32.12 ± 1.83 ^a^	31.92 ± 1.03 ^b^	19.82 ± 1.26 ^b^	8.51 ± 0.03 ^a^
Hoki	Fresh	80.52 ± 1.52 ^b^	12.13 ± 0.59 ^a^	1.90 ± 0.13 ^a^	1.40 ± 0.04 ^b^
	Deep-fried *	23.94 ± 0.85 ^a^	29.63 ± 0.78 ^b^	31.20 ± 1.06 ^b^	8.33 ± 0.06 ^a^
Prawn	Fresh	73.32 ± 1.02 ^b^	20.10 ± 0.88 ^a^	1.10 ± 0.02 ^a^	1.70 ± 0.02 ^b^
	Deep-fried *	17.97 ± 0.84 ^a^	36.84 ± 1.35 ^b^	30.61 ± 0.56 ^b^	8.25 ± 0.07 ^a^

Values are mean ± SD for triplicate replications. Mean values in the same column not sharing a common superscript under each treatment and sample are significantly different (*p* < 0.05). * Calculated from the deep-fried seafood with no added antioxidants in the frying oil.

## Data Availability

Data is contained within the article.

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
