# Peer review of "The Inhibitory Effects of Hydroxytyrosol, α-Tocopherol and Ascorbyl Palmitate on Lipid Peroxidation in Deep-Fat Fried Seafood"

_antioxidants, 2023, doi:10.3390/antiox12040929_

Round 1
Reviewer 1 Report
The work is written in a comprehensible way. However, I suggest correcting/adding some things of a formal nature:
- I have a question - in connection with the text in point 2.7. - line 142-143 - didn't the authors try to store fried seafood instead of in bags with an airtight seal in bags with filtered air (oxygen-free environment, vacuum packaging)?
- Explain “meq/kg” in POV determining (line 160)
- state the units when determining p-anisidine and acid value (lines 177 and 185)
- The Discussion chapter is missing. However, the authors discuss the results directly in Results, therefore chapter 3 should be named "Results and Discussion"
- Figure 1 and 2 – add in the name of Figure …. Sunflower oil (SFO).
- Figures 1 - 5 show the data as means +/- standard deviation. This is not an illustrative representation of the variance of the averaged values. I recommend changing the images to columns with all values marked (GraphPad Prism.)
- Table 2 – explain letters a, b
- Figure 4 -5 - use the same values on the "Y" axis - the reader can better assess the difference in results for individual seafood (for Figure 4 - Y axis 1-25, Figure 5 - Y axis 1-7 (?))
- Fill in the number of frying cycles (line 541) (although it follows from the Figure 6)
- In all figures explain in the text to the figure used abbreviations
Author Response
Response to Reviewer 1 comments:
The work is written in a comprehensible way. However, I suggest correcting/adding some things of a formal nature:
- Comment: I have a question - in connection with the text in point 2.7. - line 142-143 - didn't the authors try to store fried seafood instead of in bags with an airtight seal in bags with filtered air (oxygen-free environment, vacuum packaging)?
Response: We appreciate the reviewer’s comment. We stored the samples in an air-tight bag in order to prove that if oxidation occurred in the seafood lipids, it was not a result of the air circulation in the fridge but due to the induced oxidation resulting from the frying process. If we don’t do this, we will not be able to distinguish between the oxidation resulting from frying or from the air in the fridge.
- Comment: Explain “meq/kg” in POV determining (line 160)
Response: A further explanation has been added. Please see lines 160-161
- Comment: State the units when determining p-anisidine and acid value (lines 177 and 185)
Response: We thank the reviewer. P-anisidine results have no unit, but the unit of acid value has been included. Please, see lines 186-187
- Comment: The Discussion chapter is missing. However, the authors discuss the results directly in Results, therefore chapter 3 should be named "Results and Discussion"
Response: We thank the reviewer. We have changed the heading “Results” to “Results and Discussion”. Please see line 195
- Comment: Figure 1 and 2 – add in the name of Figure …. Sunflower oil (SFO).
Response: We thank the reviewer. We have added the missing “oil” in both Figures. Please, see lines 252 and 256
- Comment: Figures 1 - 5 show the data as means +/- standard deviation. This is not an illustrative representation of the variance of the averaged values. I recommend changing the images to columns with all values marked (GraphPad Prism.)
Response: We thank the reviewer for this comment. We know that variance helps to find the distribution of data in a population, but because of multiple data information, we decided to express our results as the mean ± standard deviation (n = 3) for clarity. This is because standard deviation gives more clarity about the deviation of data from a mean, as shown in our results. We had initially presented our results in table form, but we found that multiple columns were needed, and that complicated our ability to compare our mean values within each treatment (antioxidants, seafood samples and storage days).
- Comment: Table 2 – explain letters a, b
Response: We thank the reviewer for this comment. We mentioned under the Table that Mean values in the same column not sharing a common superscript are significantly different (P < 0.05). The superscripts are the letters a,b. However, we have added “…under each treatment and sample…” for further clarification.
- Comment: Figure 4 -5 - use the same values on the "Y" axis - the reader can better assess the difference in results for individual seafood (for Figure 4 - Y axis 1-25, Figure 5 - Y axis 1-7 (?))
Response: We thank the reviewer for this comment. These changes have now been made. Figure 4 now has Y-axis as 1-25 and Figure 5’s Y-axis now range from 1-7.
- Comment Fill in the number of frying cycles (line 541) (although it follows from the Figure 6)
Response: We thank the reviewer for this comment. We have included the total number of frying cycles achieved in this study. Please, see lines 541-542.
- Comment: In all figures explain in the text to the figure used abbreviations
Response: We thank the reviewer for this comment. We have checked through the manuscript and ensured that all abbreviated words are written in full at first mention.

Reviewer 2 Report
In the manuscript the authors examine the effects of three different antioxidants in deep frying processing and storage in a refrigerated temperature for 2 day.
The authors presented this theme based on 41 cited articles, what seem to be adequate to the specific field of the manuscript.
The presented data are interesting and the article is well described, however I have some comments.
11. the authors compare three antioxidants added to sunflower oil used for frying three seafood but do not compare with extra virgin olive oil which naturally contains hydroxytyrosol, I suggest you also try this oil for comparison.
22. Does the amount of antioxidants used in the experiments and chosen by the authors correspond to the natural amount present in olive oil?
33. Does the addition of this antioxidant and in particular of hydroxytyrosol not alter the organoleptic characteristics of the product?
44. In the conclusion paragraph authors suggest the use of oil with phenols and hydroxytyrosol for deep frying, but this is in contrast with the paper, it is obvious that EVO oil contains hydroxytyrosol and phenols naturally. Can the authors suggest also the use of seed oils added with antioxidants? would it be economically viable? What future attractions can this work have? any industrial implications? Please better describe and discuss the unique feature of this research, the aim and the possible future use of the knowledge.
55. Line 87. Delete the point before materials (paragraph 2.1)
Author Response
Response to Reviewer 2 comments:
In the manuscript the authors examine the effects of three different antioxidants in deep frying processing and storage in a refrigerated temperature for 2 day.
The authors presented this theme based on 41 cited articles, what seem to be adequate to the specific field of the manuscript.
- Comment: The authors compare three antioxidants added to sunflower oil used for frying three seafood but do not compare with extra virgin olive oil which naturally contains hydroxytyrosol, I suggest you also try this oil for comparison.
Response: We appreciate the reviewer’s comment. The reason why we did not make this comparison is because the fatty acids profiles of sunflower oil and olive oil are different. For example, their levels of saturated fatty acids such as myristic acid, palmitic acid and stearic acid are different. The higher the amount of saturated fatty acids, the more stable the oil, so our stability results will be different for both oils even if we manage to keep their levels of hydroxytyrosol the same. Also, olive oil has other antioxidant that sunflower oil does not, so their stability profiles will be different.
However, the reviewer’s suggestion exactly aligns with our current/ongoing deep-frying research for the same seafood where we are only using olive oil as the frying oil without any additional antioxidant. This will help us understand the impact of the antioxidant on the stability of the seafood lipids
- Comment: Does the amount of antioxidants used in the experiments and chosen by the authors correspond to the natural amount present in olive oil?
Response: Again, we thank the reviewer for this comment. The simple answer to this question is “no” but we have some explanations. The levels of hydroxytyrosol in olive oil brands in Australia is different, this is why we cannot use a specific amount. Our experiment was designed by using the same concentration for each of the antioxidant. This allows us to compare their effectiveness.
Again, our ongoing project is looking at different olive oil brands in Australia and analysing their antioxidant types and levels before using them to fry seafood. Our aim is to understand if these concentrations would affect the stability of the seafood lipids. We are particularly focusing on how the hydroxytyrosol in each olive oil can influence the stability of the seafood after deep-fat frying.
- Comment: Does the addition of this antioxidant and in particular of hydroxytyrosol not alter the organoleptic characteristics of the product?
Response: We appreciate the reviewer’s comment. We believe that hydroxytyrosol concentration of 1.5mmol/kg oil (maximum) used in this study is too negligible to affect the organoleptic properties of the product. However, organoleptic/sensory studies are beyond the scope of this work. These are areas that we will consider in the future
- Comment: In the conclusion paragraph authors suggest the use of oil with phenols and hydroxytyrosol for deep frying, but this is in contrast with the paper, it is obvious that EVO oil contains hydroxytyrosol and phenols naturally. Can the authors suggest also the use of seed oils added with antioxidants? would it be economically viable? What future attractions can this work have? any industrial implications? Please better describe and discuss the unique feature of this research, the aim and the possible future use of the knowledge.
Response: We thank the reviewer. We have added more texts to the manuscript to reflect the reviewer’s comments. Please see lines 622-631 “Since olive-derived polyphenols such as hydroxytyrosol and tyrosol are abundant in olive processing waste, and several studies have shown that these compounds can be cost-effectively extracted, future studies should focus on their food utilization. For instance, cheaper seed oils with added natural antioxidants such as hydroxytyrosol should be investigated further because these may be an economical way of industrial production of polyphenol-rich oils that may be cheaper than olive oil, making it available to many consumers.
This study demonstrated that natural antioxidants could be used instead of chemically synthesized ones for suppressing lipid peroxidation during deep-fat frying and refrigerated seafood storage”
- Comment: Line 87. Delete the point before materials (paragraph 2.1)
Response: We appreciate the reviewer’s comment. The point before materials has been deleted.
Round 2
Reviewer 2 Report
The authors didn't accept the suggestions but they explained the reasons, so I accept their answers.
At my point 4 they answered by adding the part to the conclusions